# Preparation of Cu/Sn-Organic Nano-Composite Catalysts for Potential Use in Hydrogen Evolution Reaction and Electrochemical Characterization

**DOI:** 10.3390/nano13050911

**Published:** 2023-02-28

**Authors:** Nezar H. Khdary, Gaber El Enany, Amani S. Almalki, Ahmed M. Alhassan, Abdullah Altamimi, Saeed Alshihri

**Affiliations:** 1Institute of Materials Science, King Abdulaziz City for Science and Technology, Riyadh 11442, Saudi Arabia; assalmalki@kacst.edu.sa (A.S.A.); amalhassan@kacst.edu.sa (A.M.A.); aalhassan@kacst.edu.sa (A.A.); alshihri@kacst.edu.sa (S.A.); 2Department of Physics, College of Science and Arts in Uglat Asugour, Qassim University, Buraydah 52571, Saudi Arabia; g.elenany@qu.edu.sa

**Keywords:** homogenous organic composite, metal-organic catalyst, electrocatalyst, hydrogen evolution, Sn, Cu

## Abstract

In this work, the solvothermal solidification method has been used to be prepared as a homogenous CuSn-organic nano-composite (CuSn-OC) to use as a catalyst for alkaline water electrolysis for cost-effective H_2_ generation. FT-IR, XRD, and SEM techniques were used to characterize the CuSn-OC which confirmed the formation of CuSn-OC with a terephthalic acid linker as well as Cu-OC and Sn-OC. The electrochemical investigation of CuSn-OC onto a glassy carbon electrode (GCE) was evaluated using the cyclic voltammetry (CV) method in 0.1 M KOH at room temperature. The thermal stability was examined using TGA methods, and the Cu-OC recorded a 91.4% weight loss after 800 °C whereas the Sn-OC and CuSn-OC recorded 16.5 and 62.4%, respectively. The results of the electroactive surface area (ECSA) were 0.5, 0.42, and 0.33 m^2^ g^−1^ for the CuSn-OC, Cu-OC, and Sn-OC, respectively, and the onset potentials for HER were −420, −900, and −430 mV vs. the RHE for the Cu-OC, Sn-OC, and CuSn-OC, respectively. LSV was used to evaluate the electrode kinetics, and the Tafel slope for the bimetallic catalyst CuSn-OC was 190 mV dec^−1^, which was less than for both the monometallic catalysts, Cu-OC and Sn-OC, while the overpotential was −0.7 vs. the RHE at a current density of −10 mA cm^−2^.

## 1. Introduction

Burning non-renewable energy sources such as fossil fuels to produce energy has several problems, such as increasing carbon dioxide emissions, which is a serious factor contributing to the greenhouse effect. It is important to recognize that global warming is closely related to using fossil fuels as energy resources, and further, one of the main energy resources, namely, crude oil, could be exhausted by the mid-21st century [1,2]. Consequently, there is an urgent need to find renewable energy resources free from environmental pollution.

Hydrogen has been referred to as the fuel of the future with water as an oxidation product with no carbon and a higher calorific value than any other chemical fuel [3,4,5]. At present, hydrogen is produced worldwide from various sources, e.g., steam reforming of methane, coal gasification, and water electrolysis. Hydrogen produced from electrocatalytic water splitting has been proven as an alternative clean and sustainable fuel to finite fossil fuels [6]. Although water splitting is a cleaner method for H_2_ synthesis, economic limitations have prevented it from playing a more significant role in H_2_ production [2,7]. Water splitting comprises two half-reactions, the oxygen evolution reaction (OER) at the anode and the hydrogen evolution reaction (HER) at the cathode. HER kinetics are significantly faster in acidic media than in alkaline media, but the process is expensive due to many factors such as electrode corrosion. However, the electrolysis of alkaline water is of paramount importance as it is one of the most widely used techniques in the industry [8], while alkaline media also enable the use of non-noble metals as catalysts at low prices [9], also provide better stability to non-noble metals by avoiding their corrosion and dissolution, hence resulting in prolonged catalysis and making it an attractive alternative to acidic catalysis [10].

In a basic solution, adsorbed H can only be produced from water molecule reduction by transferred electrons [11,12,13,14] (Volmer step):
(1)
$$H_{2}O+e⇌H_{ads}+OH^{-}\;\mathrm{Volmer}$$

followed by the recombination of adsorbed hydrogen atoms, *H_ads_*, which occurs via the Heyrovsky or the Tafel step:
(2)
$$H_{2}O+H_{ads}+e⇌H_{2}+OH^{-}\;\mathrm{Heyrovsky}$$


(3)
$$2H_{ads}⇌H2\;\mathrm{Tafel}$$


Generally, in a basic medium, a catalyst is required to break the stronger covalent H-O-H bond prior to adsorbing H, whereas the OH^−^ group in the basic medium competes with H to absorb on the active site of the catalyst. The high catalytic activity of Noble metals such as Pt for HER is attributed to their affinity for hydrogen adsorption and to facilitating the Hads recombination; however, they tend to be poor materials for cleaving the H-OH bond required for the Volmer step. On the other hand, non-noble oxophilic transition metals, while being poor catalysts for the recombination step, are quite efficient for water dissociation [15]. Therefore, moderate adsorption energies for water and hydrogen on the active sites with a low attraction to hydroxyl ions are required to proceed with a reaction in an alkaline medium at lower overpotentials and better efficiencies. Yan et al. demonstrated that the relation between the HER exchange current density in an alkaline medium and the H-binding energy values can be correlated via a volcano-type of relationship, supported both by the experimental and DFT studies [16]. It is clear that in alkaline media, Cu is located on the right branch of the volcano curve because it binds to hydrogen weak [17]. As pointed out by the Sabatier principle, neither too strong nor too weak binding would favor the overall reaction because strong or weak binding leads to either difficulty in removing the final product or poor adsorption of the reactant and this principle appears to apply in both basic and acidic electrolytes. Hydrogen and hydroxyl are important surface-bonded intermediates during many reactions in advanced energy conversion systems, in particular hydrogen evolution, hydrogen oxidation, oxygen evolution, oxygen reduction, and the oxidation of small organic molecules [18,19].

Many previous studies have been undertaken to achieve these requirements by using more than one metal compound to prepare the hybrid catalysts for HER, and most of them used transition metals (d-Blook element) [20,21,22] such as Cu, Ni, Co, and Ag. Sn, as one of the sp metals of the group four elements, has two oxidation states, namely, Sn (II) and Sn (IV), and the stability of the +2 oxidation state results from the relativistic contraction [23] of the electrons, which tends to draw the electrons closer to the nucleus than you would expect, affecting the s electrons much more than p electrons. For this reason, Sn and concentrated hydrochloric acid are traditionally used to reduce nitrobenzene to phenylamine (aniline). This reaction involves the tin first being oxidized to Sn (II) ions and then further to the preferred Sn (IV) ions. Consequently, one can conclude that the presence of Sn or Sn (II) in the catalyst might enhance the absorption and ionization of water molecules in a basic medium and could also act as a self-reducing agent for hydrogen ions.

From this point, a homogenous organic nano-composite of Cu, Sn, and bimetallic CuSn were papered by a solvothermal solidification process and thermal annealing, for investigation of its catalytic activity in HER.

## 2. Materials and Methods

### 2.1. Catalysts Preparation

Synthesis of a metal-organic nano-composite (M-OC). Typically, a known amount of a metallic compound, as shown in Table 1, was dissolved in a known volume of DMF (Dimethylformamide) with sonication, then 3 mmol of terephthalic acid C_6_H_4_(COOH)_2_ was added and the mixture was sonicated until a clear solution was observed. The resulting mixture was transferred to an autoclave reactor for 2 days at 90 °C and the obtained precipitate was then washed several times with DMF and isopropanol and then dried under a vacuum.

### 2.2. Catalysts Characterizations

The crystal structures of the samples were characterized using an X-ray diffractometer (D8 Advance, Bruker, Karlsruhe, Germany) with a Cu-Kα X-ray 2.2 kW source. Infrared absorption spectra were taken by an FT-IR spectrometer with a resolution of 4 cm^−1^ and 64 scans (Vertex 70—Bruker, Mannheim, Germany). The weight loss of the samples was measured using a TGA thermal analyzer (STD-Q600, New Castle, DE, USA) from room temperature to 800 °C at a heating rate of 10 °C/min in a nitrogen atmosphere. A scanning electron microscope (SEM) JSM-7800F (JEOL, Akishima, Japan), combined with energy dispersive X-ray spectroscopy (EDS), was performed to investigate the morphology and microstructure analysis of the prepared catalysts.

### 2.3. Electrochemical Measurements

The electrochemical characterization was performed with a multichannel potentiostat/galvanostat VSP 150 (BioLogic) connected to a computer with the EC-Lab software using three electrode configurations. Platinum mesh (Pt) and sodium saturated calomel electrodes (SSCE) were used as the counter and reference electrodes, respectively. Working electrodes were fabricated by depositing the catalyst on glassy carbon (GC) electrodes (3 mm diameter). The GC electrodes were polished using an aqueous alumina Al_2_O_3_ suspension (5 and 0.25 μm, Allied High-Tech Products Inc., Compton, CA, USA) on polishing pads. The GC electrode was then sonicated in 1 M KOH for 5 min to dissolve any embedded alumina and then rinsed with water and dried in air. To prepare the catalyst ink, 10 mg of the catalyst was dispersed in 500 μL of isopropanol, then, the ink was pipetted into a pretreated GC surface to give a loading of 1.5 μg/cm and dried under an ambient environment. The electrochemical measurements were conducted in 0.1 M KOH electrolyte at room temperature (25 ± 2 °C). The electrolyte was prepared by using KOH (≥85% KOH basis, Sigma Aldrich, MO, USA) and water (18.2 Ω·cm, Milli-Q water).

The electrolyte was purged for 30 min with N_2_ gas before usage and during the experiment to remove any dissolved gases during the electrochemical measurements. The LSV polarization curves were recorded in a potential range of −0.8 to −1.5 V vs. SSCE at a sweep rate of 10 mV/s. The electrochemically active surface area (ECSA) for each system was estimated from the electrochemical double-layer capacitance of the catalytic surface. The CV used for the electrochemical double-layer capacitance (*C_dl_*) calculation was acquired in a potential window where no faradaic process occurred from −0.2 to −0.5 V vs. the SSCE at 5, 10, 30, 50, 70, 100, and 200 mV/s, while the charging current density (*j_c_*) was captured at −0.45 V vs. the SSCE. The double-layer capacitance of the electroactive materials (*C_dl_*) was calculated from the slope of the relation between the scan rate and the charging current density. The ECSA for each system was estimated according to the relation of ESCA = *C_dl_*/C_s_, where C_s_ is the specific capacitance (0.040 mF/cm^2^) [24].

All measured potentials we reconverted to a reversible hydrogen electrode (RHE) using the following equation: E vs. RHE = E vs. SSCE + 0.236 + 0. 059 pH.

## 3. Results

### 3.1. Synthesis and Microstructural Characterization

The metal-OCs were prepared by the solvothermal solidification method using Cu (II), Sn (IV), and Cu (II)/Sn (IV) as the central metals while terephthalic acid was the linker molecule. The as-synthesized Sn-OC, Cu-OC, and CuSn-OC were obtained as light yellowish-white color, light blue, and light green powders, respectively (Figure 1). The morphology of the prepared OCs samples is presented in the SEM images in Figure 2, and it is obvious that the method of synthesis led to three different shapes, namely, spherical with around 300 nm in the case of the Sn-OC, cubic-like in the case of the Cu-OC, and flower-like in the case of the CuSn-OC with an open structure. The EDX data, (Table 2) illustrates the weight and atomic percentages of each element, and the ratio between the Cu to Sn was around 3:1. This result reflects the high susceptibility of copper to chelating with terephthalic acid, as the proportion of copper in the resulting compound was three times that of tin.

Terephthalic acid had a crystalline nature as maintained in file no. (311916, 211919) at the ICCD, but when Sn was reacted with terephthalic acid, an amorphous complex was formed as Figure 3c shows. Figure 3b shows different behavior when Cu reacted with terephthalic acid, where a crystalline complex of Cu–terephthalic acid was formed, (see file no. (381629) at the ICCD), revealing the existence of strange peaks between 2θ angles (10–16 degrees), marked by (*), which may refer to the formation of Cu(OH)_2_.H_2_O (see file no. (420746) at the ICCD). Figure 3a shows a strong peak at 2θ = 19°, which confirms the formation of a complex between Cu and terephthalic acid. Additionally, Figure 3a does not deny the formation of an amorphous complex between Sn and terephthalic acid because the Sn and terephthalic acid formed an amorphous complex as mentioned at the beginning of this paragraph; therefore, we can expect the presence of two phases in this state where one is crystalline with Cu and the other with Sn.

According to the Debye–Scherrer formula, the average particle sizes D of the samples were given by: [25,26] D_hkl_ = 0.9λ⁄βcosθ, whereby β is the full width at a half-maximum (FWHM) value of the XRD diffraction lines (see Figure 3a,b), λ is the wavelength whose value was 0.154056 nm, and θ is the half diffraction angle of 2θ. The crystalline sizes were determined at the highest peaks at ca. 2θ = 17.18° and 19.23°, and this is marked by (▼) in both Figure 3a,b. They were 24.34 and 62.68 nm for the Cu-OC and CuSn-OC, respectively.

The microstructure of the prepared samples was further investigated by FTIR spectroscopy as shown in Figure 4. It was found that at high-frequency regions, the nearly identical broad band located from 2000 to 3500 cm^−1^ for both Sn-OC and CuSn-OC samples may have been assigned to those carboxylate groups located in the big pores. Water may have been penetrating the carboxylic acid groups only in the large pores, where it protonated the carboxylate groups, and signals around 1700 and 1200 cm^−1^, characteristic of carboxylic acid groups, appeared [27]. These results reflect the affinity of the sample containing the Sn cation to absorb more water during the washing process, and the result was supported by the results of the TGA in the next section. Additionally, the broadband may be explained in the base of the formation of a dimeric form of terephthalic acid, which gave rise to a broad absorption region with many sub-maxims [28] at 2500–3000 cm^−1^. In the region of 2500–3500 cm^−1^, we would expect also to find the C-H stretching modes of the benzyl group. The band at 578 (Figure 3c), 573 (Figure 3b), and 548 cm^−1^ (Figure 3a), indicate the presence of a M-O bond in the Sn-OC, Cu-OC, and CuSn-OC, respectively.

The band at about 3509 cm^−1^ for the carboxylic acid virtually disappeared from the spectra of the prepared OC samples. The OC samples exhibited a more broadened band in the region near 2970–2980 cm^−1^, indicating the presence of coordinated water molecules. The coordinated water in all the oligomeric metal (II) complexes presented different peaks at 1000 cm^−1^ (rocking) and 750 cm^−1^ (wagging), whereas none of these vibrations appeared in the spectra of uncoordinated ligands [29].

The thermal behavior was investigated by a TGA thermal analyzer at a heating rate of 10 °C min^−1^ in the temperature range of 50–800 °C under nitrogen. The Cu-OC lost its weight in four stages, with the initial weight loss occurring in the range of 30 to 170 °C due to the evaporation of moisture content, and the total mass loss was about (9.66%). In the second and third steps, the weight loss where the ca. was 84%, which occurred between 170 and 440 °C, exhibited a mass loss corresponding to the loss of coordinated water molecules to the metal ion and decomposition of the ligand, as shown in Figure 5a. The decomposition of the Sn-OC, shown in Figure 5b, occurred in four steps as in the case of the Cu-OC. The first step occurred between 30 and 110 °C, which might be attributed to the mass loss corresponding to the absorbed water molecules. In steps two and three, a very low weight loss occurred in the range of 110 to 450 °C, where an observed ca. of 8% reflected the low loss of coordinated water molecules and good thermal stability for the obtained sample with a total mass loss of 16%. In the case of the CuSn-OC, the three steps of decomposition were observed, as shown in Figure 5c, and the presence of a Sn ion in the sample made it more thermally stable than in the case of using a Cu ion only, where the total mass loss was 26%.

### 3.2. Electrochemical Properties

#### 3.2.1. Cyclic Voltammetric Studies

For the Cu-OC, as shown in Figure 6a, a large diminishing of the peek current appeared, and oxides of copper were produced with limited solubility as the potential was increased in the case of alkali metal hydroxide solutions [30]. Additionally, soluble species such as Cu(OH)_4_^−2^ were also speculated to form [31], which explained the large decay in the peak current. In Figure 6b, peak 1 may have been attributed to the formation of Cu_2_O whereas peak 2 may have been attributed to the soluble product of Cu(OH)_4_^−2^. The two peaks in the reduction region may have been due to the formation of Cu from CuO, at a potential of −0.12(V) and Cu(OH)_4_^−2^, or at a more negative optional up to −1 (V). As seen in Figure 6c, the anodic peak current increased linearly with an increasing scan rate, which indicates that the redox process was non-diffusional.

For the cyclic voltammograms of the Sn-OC electrode in 0.1 M KOH at scan rates of 200 mVs^−1^, as represented in Figure 7a, it can be observed that peak S was only observed in a fresh sample on the first scan cycle, then disappeared with successive scans and was not observed again. This peak may have resulted from the reduction of Sn IV to Sn (II) or Sn particles. Peak S3, which appeared in the second cycle, could be attributed to the oxidation of metallic tin to Sn (II), probably forming Sn(OH)_2_ or SnO, whereas peak S4 may be attributed to the oxidation of the previously formed Sn (II), giving Sn(OH)_4_. The passivation region after peak S4 has been attributed to the progressive conversion of Sn(OH)_4_ to SnO_2_. At the reduction region, peaks S1 and S2 were associated with the simultaneous reduction of the Sn(IV) and Sn(II) to Sn particles. Generally, by increasing the cycle number the oxidation charge becomes much more than the reduction one, which indicates the formation of passive oxides or hydroxides, and this results in a decrease in hydrogen and oxygen evolution. At cycle 25, peaks S3 and S4 diminished, and small broadband formed, as shown in Figure 7b. The peak current for peaks S2 and S5 increased linearly with an increasing scan rate, as shown in Figure 7c, indicating that the redox process was non-diffusional, whereas the peak current for peak S1 increased linearly with increasing the square root of the scan rate, indicating that the redox process was diffusional, as shown in Figure 7d.

For the CuSn-OC, Figure 8 represents the response of the CuSn-OC sample in a 0.1 M KOH solution at a scan rate of 200 mV/s, where the behavior of the first reduction peaks may be attributed to the transition of Sn (IV) to Sn(II), and then to the Sn metal and deposition of Cu to form a Sn-Cu bimetal onto the surface of the GC substrate. In the oxidation region, the dissolution of the Cu and Sn was started and the formation of a thin layer from passive SnO_2_ may have been formed. The decrease in the peak’s current and charge in the next cycles may have been due to this layer.

#### 3.2.2. HER Performance Measurements

The HER performances of the Cu-OC, Sn-OC, and CuSn-OC catalysts were evaluated in a 0.1 M KOH electrolyte via a conventional three-electrode system. Linear sweep voltammetry, as shown in Figure 9 for the prepared catalyst samples, revealed that the overpotential at −10 mA cm^−2^ was in the order of Sn-OC > CuSn-OC > CuSn-OC, where the Cu-OC required a lower overpotential than the CuSn-OC of only −30 mV at −10 mA cm^−2^. The overpotential values at −10 mA cm^−2^ are shown in Table 3. We have taken the onset potential from Figure 9 and the values were −420, −900, and −440 mV vs. the RHE for the Cu-OC, Sn-OC, and CuSn-OC, respectively. In the case of the Sn-OC, the d-band of the Sn sp metal, would be expected to lie low, thus it would play no role in the bonding of hydrogen nor electrocatalysis [32].

The difficulty with Sn-OC is that measurable currents can only be obtained at high overpotentials (i.e., an onset potential = −900 mV), therefore, the determination of the exchange current density *j_o_* requires an extrapolation over a large potential range. The Fermi level of the d metals lies within the d band as a result, and the hydrogen adsorption ability on the Cu surface is more than that on the Sn surface; therefore, the onset potential of the catalysts containing Cu, Cu-OC, and CuSn-OC, were much less than in the case of the Sn-OC, as shown in Table 3.

The electrochemically-active surface area (ECSA) for each system was estimated from the electrochemical double-layer capacitance of the catalytic surface (Figure 10a), as shown in Table 3. The result reflects more ECSA for the bimetal ions catalyst of CuSn-OC, with 0.5 m^2^ g^−1^, compared with the monometallic catalysts Cu-OC and CuSn-OC.

The HER process in an alkaline medium is governed by the dissociation of water molecules, where two intermediate species form, namely, OH^−^ and H^+^, which need to adsorb and desorb on the catalyst surface. Adsorbed H, in an alkaline medium, can only be produced from water molecule reduction by transferred electrons (Volmer step), and the catalyst requires the breaking of a stronger covalent H-O-H bond prior to adsorbing H, taking into consideration the fact that hydroxide ions present in a solution tend to adsorb onto a catalyst surface, thereby competing with hydrogen and reducing the number of available adsorption sites needed for HER progress [17].

To evaluate the HER kinetics of these electrocatalysts, the Tafel slope was calculated and plotted as shown in Figure 10b. the Tafel slope of the CuSn-OC was 195 mV dec^−1^, which was lower than for both the Cu-OC at 335 mV dec^−1^, and the Sn-OC at 250 mV dec^−1^, indicating the synergetic effect of Cu and Sn ions in its binary catalyst. The exchange current density, *j_o_*, and the transfer coefficient, α, were also estimated as shown in Table 3.

The Tafel slope in the case of a catalyst containing Sn was less than that of a Cu-OC catalyst, even though a catalyst containing Cu had a higher ability to absorb hydrogen than tin, and an ECSA of Cu-OC was more than for Sn-OC. This indicates that the low bonding of hydrogen in the Sn-OC catalyst did not imply that the interaction of the d band with the adsorbed hydrogen was weak. The bimetal ions catalyst CuSn-OC had a lower Tafel slope value and higher ECSE; therefore, the CuSn-OC catalyst had much more catalytic activity for HER in the basic medium than both the Cu-OC and Sn-OC. This caused us to assume that the Cu form in the catalyst CuSn-OC enhanced the dissociation of water molecules and broke the stronger covalent H-O-H bond prior to adsorbing H, whereas the Sn form facilitated the reduction of the formed H^+^ to weakly adsorb H.

## 4. Conclusions

In summary, OCs of Sn, Cu, and dual SnCu were successfully obtained via solvothermal solidification using Cu (II), Sn (IV), and Cu(II)/Sn(IV) as the central metals while terephthalic acid was used as the OCs’ linker molecule. Despite the good catalytic activity of the Cu-OC for HER in an alkaline medium, it had poor thermal stability; therefore, incorporating Sn (II) ions with a Cu (II) ion to form bimetal ion OCs, namely, CuSn-OC, enhanced the thermal stability of the prepared catalyst, as shown in the TGA data. The Tafel slope of the CuSn-OC was 195 mV dec^−1^, which was lower than for both the Cu-OC at 335 mV dec^−1^, and the Sn-OC at 250 mV dec^−1^, indicating the synergetic effect of Cu and Sn in its binary metal ion catalyst.

## Figures and Tables

**Figure 1 nanomaterials-13-00911-f001:**
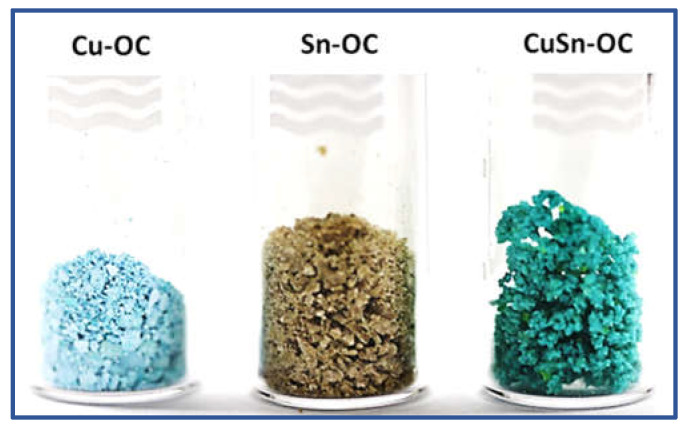
Photo images of Cu-OC, Sn-OC, and CuSn-OC.

**Figure 2 nanomaterials-13-00911-f002:**
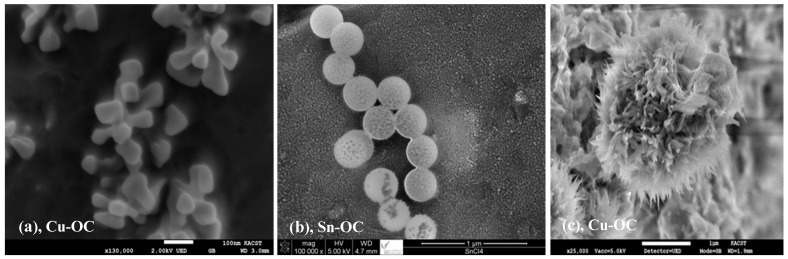
SEM images of as-prepared Sn-OC (**a**), Cu-OC (**b**), and CuSn-OC (**c**).

**Figure 3 nanomaterials-13-00911-f003:**
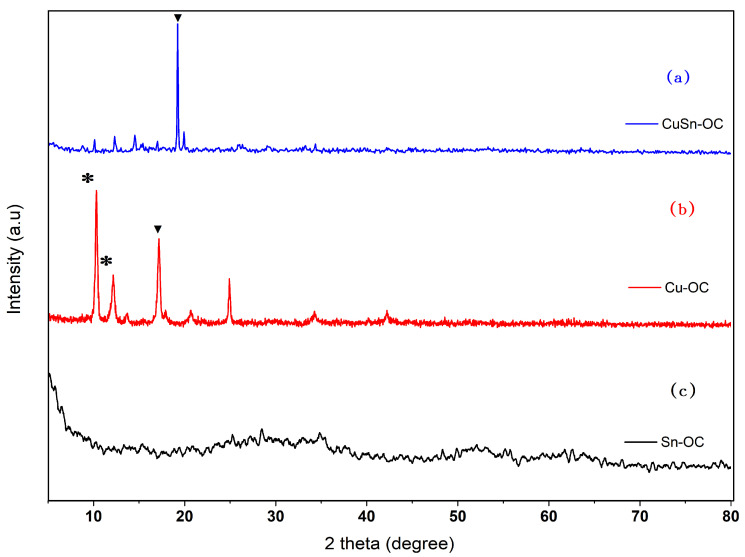
XRD patterns of (**a**) Cu-OC, (**b**) Sn-OC, and (**c**) CuSn-OC.

**Figure 4 nanomaterials-13-00911-f004:**
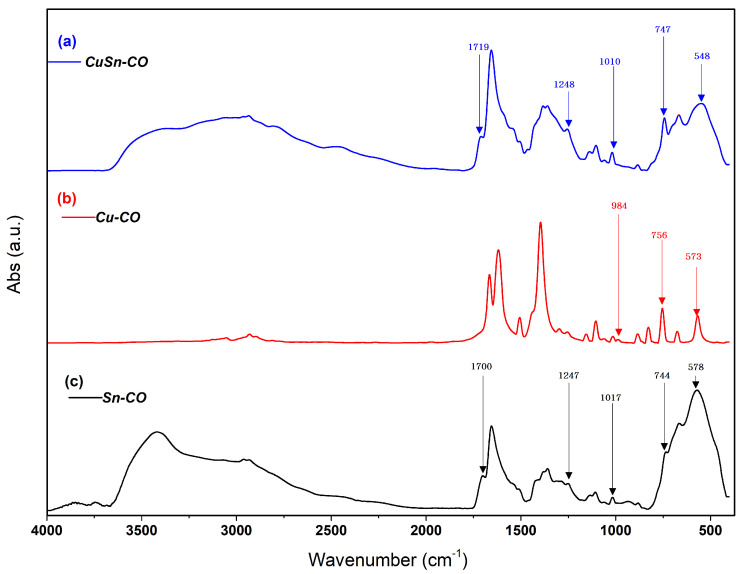
FTIR spectra of as-synthesized (**a**) Cu-OC, (**b**) Sn-OC, and (**c**) CuSn-OC.

**Figure 5 nanomaterials-13-00911-f005:**
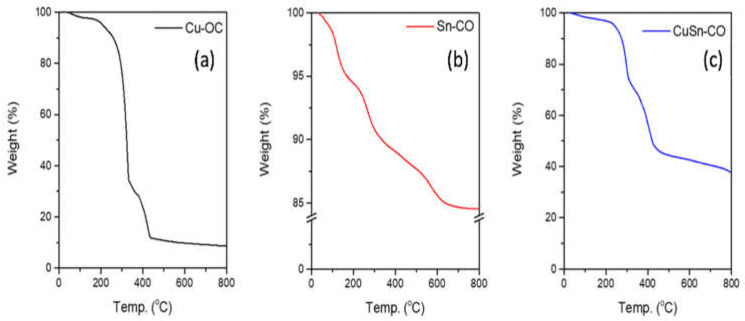
Thermogravimetric analyses for as-synthesized (**a**) Cu-OC, (**b**) Sn-OC, and (**c**) CuSn-OC.

**Figure 6 nanomaterials-13-00911-f006:**
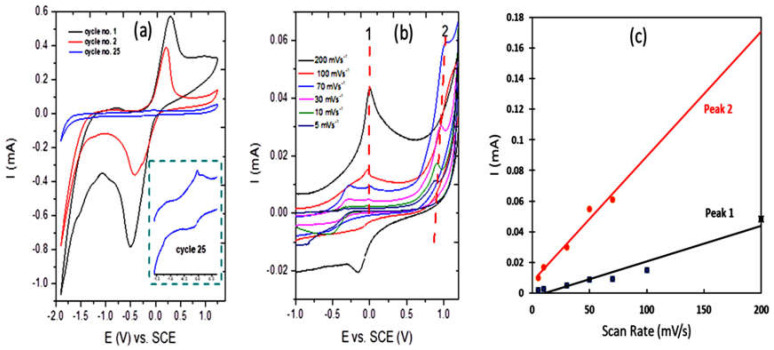
(**a**) Cyclic voltammograms of Cu-OC catalyst in 0.1 M KOH at scan rates of 200 mV/s, (**b**) cyclic voltammograms of Cu-OC catalyst in 0.1 M KOH at scan rates of 200, 100, 70, 30, 10 and 5 mV/s, and (**c**) anodic peak currents (Ia) as a function of the scan rate for the Cu-OC catalyst in 0.1 M KOH.

**Figure 7 nanomaterials-13-00911-f007:**
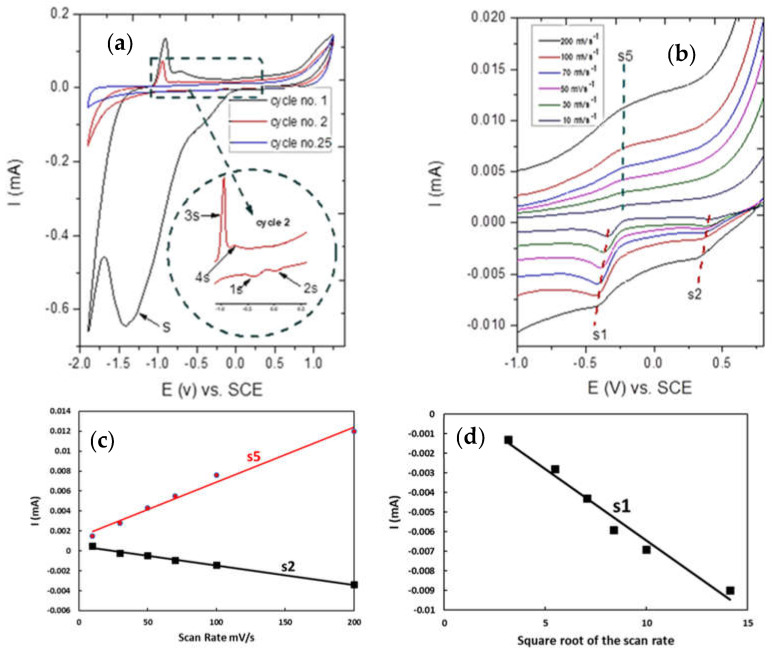
(**a**) Cyclic voltammograms of Sn-OC electrode in 0.1 M KOH at scan rates of 200 mV/s, (**b**) cyclic voltammograms of Sn-OC electrode in 0.1 M KOH at scan rates of 200, 100, 70, 50, 30 and 10 mV/s, (**c**) anodic and cathodic peak currents for Sn-OC electrode in 0.1 M KOH, as a function of the scan rate, and (**d**) as a square root of the scan rate.

**Figure 8 nanomaterials-13-00911-f008:**
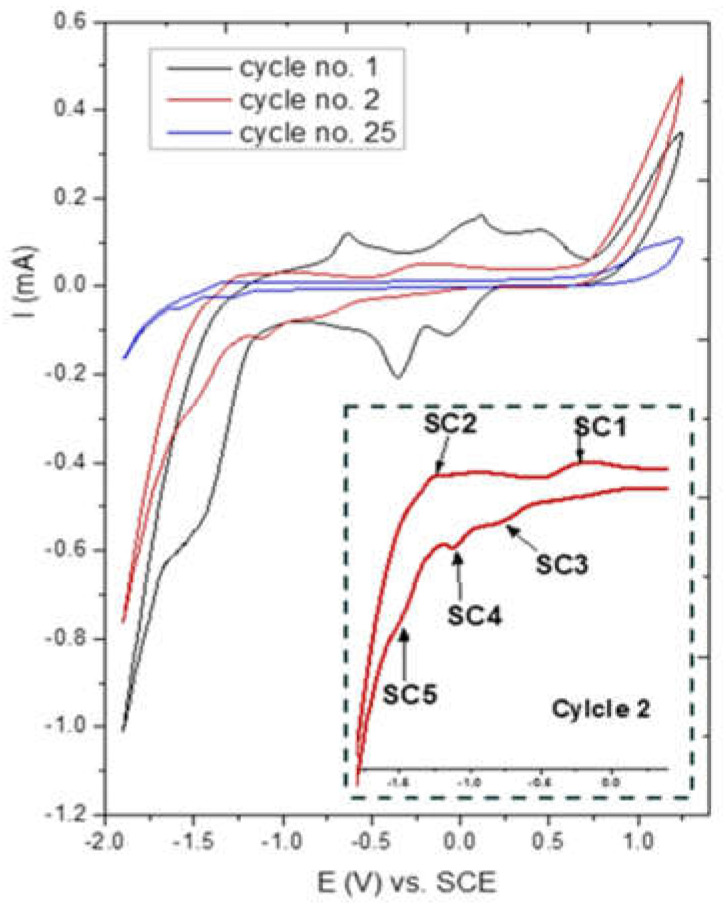
Cyclic voltammograms of CuSn-OC electrode in 0.1 M KOH at scan rates of 200 mV/s.

**Figure 9 nanomaterials-13-00911-f009:**
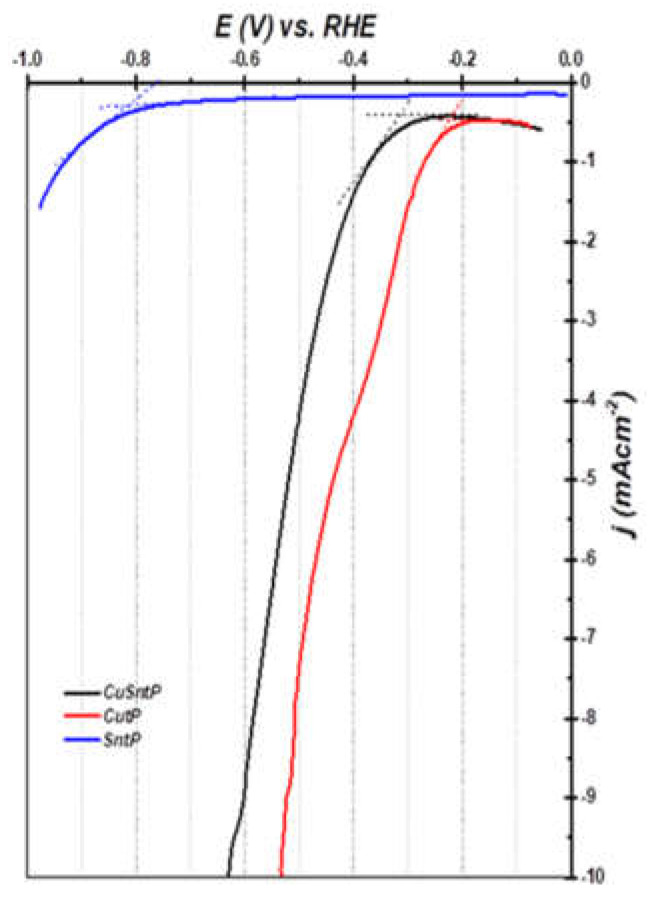
HER polarization curves of MOF catalysts in 0.1 M KOH.

**Figure 10 nanomaterials-13-00911-f010:**
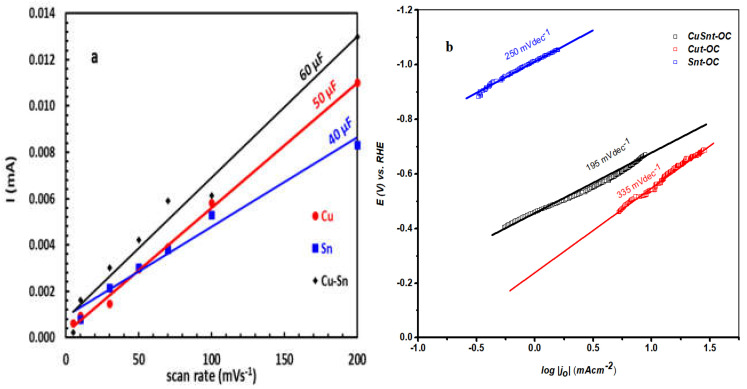
(**a**) Anodic charging current measured at the potentials of 0.5 V vs. RHE plotted as a function of the scan rate and the slope of the linear fit giving the C_dl_. (**b**) Tafel analysis of the catalysts in 0.1 M KOH.

**Table 1 nanomaterials-13-00911-t001:** The catalyst composition with the reaction components.

Catalyst	Ingredient	Formula	mmol	V (mL), DMF	Catalyst Code
Cu-OC	Cupric acetate	Cu_2_(CH_3_COO)_4_	2.75	50	Cu-OC
Sn-OC	Tin (IV) chloride	SnCl_4_	1.923	50	Sn-OC
CuSn-OC	Cupric (II) acetate monohydrate	Cu(CH_3_COO)_2_.H_2_O	2.75	100	CuSn-OC
	Tin (IV) chloride	SnCl_4_	1.923		

**Table 3 nanomaterials-13-00911-t003:** HER performance of Cu-OC, Sn-OC, and CuSn-OC catalysts in 0.1 KOH alkaline solution.

Electrocatalyst	Onset Potential (mV) vs. RHE	C_dl_ (μF cm^−2^)	ECSA (m^2^ g^−1^)	Tafel Slope mVdec^−1^	j_o_ mA/cm^2^	α	V (mV) vs. RHE at −10 mA/cm^2^
Cu-OC	−320	50	0.41	335	2.1 × 10^−1^	0.18	−630
Sn-OC	−760	40	0.33	250	3.1 × 10^−5^	0.24	--
CuSn-OC	−250	60	0.50	195	6.3557 × 10^−3^	0.31	−530

**Table 2 nanomaterials-13-00911-t002:** EDX element analysis table for CuSn-OC.

Element, Orbit	Weight %	Atomic %
C, K	23.58	41.50
O, K	31.24	41.27
S, K	9.74	6.42
Cu, K	24.95	8.30
Sn, L	8.28	1.47

## Data Availability

All data are available in electronic version as well as in the storage unit of each analysis device.

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
