# Peer review of "Preparation of Cu/Sn-Organic Nano-Composite Catalysts for Potential Use in Hydrogen Evolution Reaction and Electrochemical Characterization"

_nanomaterials, 2023, doi:10.3390/nano13050911_

Round 1

Reviewer 1 Report

The manuscript presents the solvothermal synthesis of three potential hydrogen evolution reaction (HER) catalysts, based on metal-organic composites of Cu(II) and Sn(IV) and terephtalic acid, with the aim to provide a detailed investigation on the synergistic effects of the presence of both metal ions on the composites thermal stability and HER performance.

On the positive side - the  manuscript has a relatively straightforward scientific idea, and it is obvious that the authors have attempted to collect enough data to present a detailed investigation on the effects of the Cu/Sn combination on the structure, morphology and electrochemical behaviour of the composites, however, I cannot recommend its publication in Nanomaterials in its current state, as it has some deficiencies in its structure, clarity of presentation, and discussion of the results, which makes it difficult to clearly judge the papers potential scientific merit. I believe that it should be thoroughly revised to provide more discussion, especially within the preparation and characterisation sections. Also - in the discussion of the electrochemical behaviour of the composites I do not see much comparison with results on similar HER catalysts, already published in literature, so that the authors attempt to outline the benefits and drawbacks of their proposed system.

The overall presentation quality of the manuscript should be improved. Regarding the quality of the English presentation - it is satisfactory, to some extent, however, the text is riddled with typos and misused expressions, such as “Defiant volume” in line 91; “prepend catalyst” line 111; “H2”, “SnO2”, “Sn(OH)2” written w/o the number subscripted on multiple occasions in the text.; 

I have listed some suggestions for a more detailed corrections, that would be suitable, below:

(1) The Introduction is well written and comprehensive, however, it it is poor on references and there are many statements for which no reference is provided. I suggest that the authors: (i) provide more references on the steps outlined between lines 50 - 55; (ii) do a careful read-through to correct for  typos and missexpressions - e.g., excreted alkaline water” in line 44; etc. “lower prizes” in line 45; “; (iii) add the corresponding references in the paragraph 76-85; and (iv) expand the final sentence in lines 86-88 to provide a proper introduction to the experimental part of the paper; Finally, I personally find it a bit unnerving that in the last paragraph there is a mix of nomenclatures when oxidation states are presented “tin (IV)” vs. “Sn (IV)” and that the + in “+2” is in superscript.

(2) Please correct the subsection numbering, as both Sections 2, and 3 have unnumbered subsections. It is not a big issue, but it it is a bit distracting for a reader, due to the larger font used w/o a clear indication that a new subsection is beginning. 

(3) It is not immediately clear that the Metal-OC notation stands for metal-organic composite (probably, since in the abstract is described as metal-organic-nano-composite). It should be clearly introduced in the text prior the first use (which, I believe is in line 90).

(3) In Table (1) the formula of Cupric acetate is shown as Cu2(CH3COO)4 and is correct, for the hydrated crystalline salt Cu2(CH3COO)4.2H2O, but not for the anhydrous or dissolved salt - please indicate in Section 2 whether anhydrous or hydrated salt was used to avoid confusion. 

(4) The ECSA results are still not introduced, but in the beginning of Section 3 there is already discussion on its relation with the morphology of the OCs (lines 147-149). This sentenced needs to be removed.

(5) The scale markers in Fig. 2 are misleading. Especially in Fig 2c it is clear that the original one shows 1 μm, but the authors have overlayed a second one, indicating 0.1 μm. This figure should be improved. Also, table 2 would have been more informative if a comparison of the data, obtained by Cu-OC and Sn-OC was also given, as a reference.  

(6) The discussion on the XRD data (lines 168-180) is lacking clarity and needs to be re-written: (i) the text (lines 174-175) refers to Fig. 2a and 2b, whereas the XRD data is shown in Fig. 3; (ii) Lines 178-179 state that the peaks at 17 and 25 degrees 2Theta “correspond to (511) and (731) crystal planes …”, however it is not clear of what ? The authors should clearly state which crystallographic phase do they expect to have obtained, as a result of hydrothermal synthesis and present it to the reader, since currently the only mention in the text is regarding CuO and this leads to a confusion. Figure 3 should also be updated, if possible and labelled with the corresponding reflection Miller indices (and in it it is unclear also what is indicated as “1” with the arrow in (a) and (b).

(7) The FTIR discussion is a bit superficial, although it is not an issue. However, a good reference spectra in Fig. 4 would have been the terephtalic acid linker itself. Finally, I do not see a reason to present the wavenumber with a precision of 5 characters, after the decimal point. 

(8) In the electrochemical section, lines 262-264 state that “… the process is non-diffusional and the Nafion matrix may facilitate ion mobility …”. However, nowhere in the text it is stated that Nafion was used in any way in the preparation of the catalyst ink, or as an element of the electrochemical measurements setup. May the authors clarify this ? 

(9) The panes in Fig. 7 are not labeled accordingly, with respect to the figure caption. Figure 10, as well, and also the Tafel plot pane there is missing the axis labels.

(10) The paragraphs between line 320-337 point to Table 1, however, the results are outlined in Table 2.

Author Response

Thank you for Reviewer 1. The reply is attched 

Reviewer 2 Report

This manuscript by Khdary et al. reports their experimental results on Cu/Sn organic composites as catalysts for hydrogen evolution reactions. They claimed that Cu-organic composite, Sn-organic composite, and Cu-Sn-organic composite with terephthalic linker were prepared and characterized with SEM, TGA, and IR. Their performances were evaluated with electrochemical analysis. Overall, the results are interesting for the readership of Nanomaterials, I would like to recommend publication after the following concerns were clarified:

1, The authors should interpret the IR spectra carefully, which in my opinion, is the key evidence to prove that their composites were indeed coordinated with the terephthalic linker, not the CH3COO-.

2, The reason for the imbalance of Cu/Sn contents in the Cu-Sn-organic composite deserves deeper interpretation and discussion.

Author Response

Many thanks to Reviewer 2. The reply is attached.

Kind regards

Round 2

Reviewer 1 Report

I would like to thank to the authors for the efforts in improving the quality of the manuscript and believe that it is ready to be pushed for publication in its current state. Finally, I would like to advise the authors to carefully read the fina version, and make any additional alterations and adjustments needed to its clarity and style of presentation.